Structural modeling of the flagellum MS ring protein FliF reveals similarities to the type III secretion system and sporulation complex

Bergeron Julien R. jrcberge@uw.edu
Department of Biochemistry, University of Washington , Seattle, WA , USA
Blocker Ariel
Electronic publication date: 2016 Feb 22
Publication date: 2016
Volume: 4
Electronic Location ID: e1718
Received 2015 Nov 6; Accepted 2016 Jan 31
Copyright: ©2016 Bergeron
Copyright year: 2016
Copyright holder: Bergeron
License: This is an open access article distributed under the terms of the Creative Commons Attribution License, which permits unrestricted use, distribution, reproduction and adaptation in any medium and for any purpose provided that it is properly attributed. For attribution, the original author(s), title, publication source (PeerJ) and either DOI or URL of the article must be cited.
License URL: https://creativecommons.org/licenses/by/4.0/

Keywords: Flagellum, Secretion system, Salmonella, Chlamydia, Sporulation, Homology modeling

Funding: UBC Centre for Blood Research post-doctoral transition award This work was supported by a UBC Centre for Blood Research post-doctoral transition award. The funders had no role in study design, data collection and analysis, decision to publish, or preparation of the manuscript.

==============================
The flagellum is a large proteinaceous organelle found at the surface of many bacteria, whose primary role is to allow motility through the rotation of a long extracellular filament. It is an essential virulence factor in many pathogenic species, and is also a priming component in the formation of antibiotic-resistant biofilms. The flagellum consists of the export apparatus on the cytosolic side; the basal body and rotor, spanning the bacterial membrane(s) and periplasm; and the hook-filament, that protrudes away from the bacterial surface. Formation of the basal body MS ring region, constituted of multiple copies of the protein FliF, is one of the initial steps of flagellum assembly. However, the precise architecture of FliF is poorly understood. Here, I report a bioinformatics analysis of the FliF sequence from various bacterial species, suggesting that its periplasmic region is composed of three globular domains. The first two are homologous to that of the type III secretion system injectisome proteins SctJ, and the third possesses a similar fold to that of the sporulation complex component SpoIIIAG. I also describe that Chlamydia possesses an unusual FliF protein, lacking part of the SctJ homology domain and the SpoIIIAG-like domain, and fused to the rotor component FliG at its C-terminus. Finally, I have combined the sequence analysis of FliF with the EM map of the MS ring, to propose the first atomic model for the FliF oligomer, suggesting that FliF is structurally akin to a fusion of the two injectisome components SctJ and SctD. These results further define the relationship between the flagellum, injectisome and sporulation complex, and will facilitate future structural characterization of the flagellum basal body.

Introduction

Bacteria interact with their environment using a range of surface appendages, including flagella, pili, fimbriae, and secretion systems (Fronzes, Remaut & Waksman, 2008). In particular, the flagellum is responsible for motility in many bacteria (Terashima, Kojima & Homma, 2008), but it is also frequently associated with adhesion to surfaces and/or other cells (Belas, 2014). Flagella are found in many bacterial families, including most gram-positive, proteobacteria and spirochetes (Chen et al., 2011; Minamino & Imada, 2015). Notably, it is an essential virulence factor in many pathogenic species, such as Salmonella, E. coli, Clostridium, Pseudomonas, Helicobacter, Vibrio, Burkholderia, and Campylobacter, making the flagellum a potential target for new antibacterial therapeutics (Duan et al., 2013).

The bacterial flagellum is constituted of four distinct regions (Stock, Namba & Lee, 2012; Morimoto & Minamino, 2014; Minamino & Imada, 2015) (Fig. 1A). On the cytosolic side and anchored to the inner-membrane, the type III secretion system (T3SS) apparatus is responsible for the secretion of the extracellular components. The membrane-embedded structure that traverses the cytoplasmic membrane and periplasmic space (as well as the outer membrane for gram-negative bacteria) is called the basal body, and is linked to the cytoplasmic C-ring to form the rotor complex. The hook is a curved filament, ∼55 nm in length (for the prototypical Salmonella typhimurium flagellum (Hirano et al., 1994)), that protrudes away from the basal body. It is prolonged by the filament, a long structure (>10 µm in S. typhimurium) responsible for motility and adherence.

Figure 1 The flagellar, T3SS, and sporulation complexes.

Schematic representation of the bacterial flagellum (A), the T3SS (B), and the sporulation complex (C). SctJ-like components are in blue, SctD-like components in green, and outer-membrane components in yellow. The EM maps are shown in grey for (A) and (B). The ring structures identified in the flagellum are also indicated. IM, inner membrane; OM, outer membrane; IFM, inner forespore membrane; OFM, outer forespore membrane.

Genetic studies have revealed that the inner-membrane protein FliF forms a two ring-shaped structures called the MS rings, that assembles early during flagellum morphogenesis (Kubori et al., 1992), likely around elements of the T3SS. This in turn recruits the C-ring and ATPase in the cytosol, leading to a secretion-competent complex, which can export the hook protein (and other flagellar components) to the extracellular environment through its central pore. A subsequent substrate specificity switching event, controlled by elements of the T3SS, leads to the assembly of the filament (Minamino, Imada & Namba, 2008).

FliF is an ∼60 kDa protein, localized to the inner-membrane through the Sec pathway. Sequence analysis has indicated that it possesses two transmembrane helices, flanking a large periplasmic region (Ueno, Oosawa & Aizawa, 1994). At the C-terminus, a cytosolic peptide has been shown to interact with FliG (Levenson, Zhou & Dahlquist, 2012; Ogawa et al., 2015), a component of the flagellum rotor. EM reconstructions of purified FliF revealed a homo 26-mer forming the MS ring oligomer (Suzuki, Yonekura & Namba, 2004), although analyses of intact flagellum particles have identified 24-, 25- and 26-fold symmetry for this region of the basal body (Thomas et al., 2006).

A number of components from the cytosolic export apparatus are homologous to that of the injectisome (Fig. 1B), another bacterial complex whose role is to inject so-called “effector” proteins inside the cytosol of target or symbiotic cells (Buttner, 2012; Diepold & Armitage, 2015). Indeed, phylogenetic studies have revealed that the flagellum export apparatus is likely the evolutionary ancestor of the injectisome (Abby & Rocha, 2012), with both complexes employing a similar T3SS for protein export and coordinated assembly (Buttner, 2012; Diepold & Armitage, 2015).

In particular, FliF shows significantly sequence similarity to an inner-membrane component of the injectisome, SctJ (24% sequence identity with the Salmonella homologue, 22% sequence identity with the EPEC homologue, for residues 52–217 of FliF) (Ueno, Oosawa & Aizawa, 1994). SctJ forms a 24-mer ring structure in the inner-membrane, similar to that of FliF, and structural characterization have revealed the molecular details of its architecture and oligomerization (Yip et al., 2005; Bergeron et al., 2015). Specifically, the periplasmic region of SctJ consists of two globular domains with a canonical “ring building motif” (RBM) fold (Crepin et al., 2005; Yip et al., 2005; Spreter et al., 2009), found in several oligomeric proteins. The two RBMs are joined by a rigid linker, which was shown to promote oligomerization (Bergeron et al., 2015).

Recently, two proteins essential for the sporulation process in Bacillus subtilis, SpoIIIAH and SpoIIIAG, were shown to be homologous to SctJ and FliF (Camp & Losick, 2008; Meisner et al., 2008). This led to the suggestion that these proteins are part of a complex that directs the transport of proteins and/or nutrients between the mother cell and the endospore (Fig. 1C), although such complex has not been observed directly (Crawshaw et al., 2014).

There is currently very little data on the architecture of FliF, and its relationship to the injectisome component SctJ and the sporulation complex components SpoIIIAH and SpoIIIAG. In this study, I have exploited the recent structural studies of the injectisome and sporulation complex, in order to update our understanding of the FliF architecture using computational and bioinformatics analyses. Based on these, I propose that the periplasmic region of FliF consists of a SctJ homology domain, as well as a FliF-specific domain structurally homologous to the sporulation complex component SpoIIIAG. I also report that in the Chlamydiacae family, the FliF protein differs significantly from other species, as it lacks part of the SctJ homology region, and is fused to a FliG-like domain at the C-terminal cytosolic end. Finally, I have combined previously determined EM maps and structural modeling to introduce the first molecular model for the FliF periplasmic region, suggesting that FliF is akin to a fusion of the injectisome basal body inner-membrane components SctJ and SctD. This unexpected observation has implications in the understanding of the evolutionary relationship between the flagellum, injectisome and sporulation complex.

Materials and Methods

Sequence mining, analysis and alignment

All protein sequences were identified in the National Centre for Biotechnology Information protein database RefSeq (Pruitt et al., 2012). Multiple sequence alignments were generated with ClustalW (McWilliam et al., 2013) using default parameters. For secondary structure-based multiple alignments, a composite alignment was generated manually based on the individual pairwise alignments. Alignment figures were produced with ESPript (Gouet, Robert & Courcelle, 2003).

Secondary structure elements, signal sequences, transmembrane helices and structural homologues were predicted with the PSIPRED server (Buchan et al., 2013). Protein sequence identity was calculated with the Needleman-Wunsch algorithm on the EBI server (Li et al., 2015), using default parameters. Signal sequences were predicted with SignalP 4.1 (Petersen et al., 2011).

Modeling and refinement

Structure-based alignment and initial models were obtained with Phyre (Kelley et al., 2015), and the models were further refined by performing 1,000 cycles of the Relax procedure (Rohl et al., 2004) in Rosetta 3.4 (Leaver-Fay et al., 2011), using the following flags:

– database ∼rosetta/rosetta_database

– in:file:s input.pdb

– in:file:fullatom

– relax:thorough

– nstruct 1000

– out:file:silent relax.silent

The geometry of the obtained models was analyzed with the PSVS suite (Bhattacharya, Tejero & Montelione, 2007).

EM map docking and symmetry modeling

The flagellum, injectisome, and FliF EM maps (EMDB ID 1887, 1875 and personal communication from K Namba), were docked with the MatchMaker tool in Chimera (Goddard, Huang & Ferrin, 2007). Models of the FliF RBMs were placed in their putative location of the FliF EM map manually using Chimera.

For the EM-guided symmetry modeling, 24-fold, 25-fold and 26-fold symmetry definition files were generated, and used for the rigid-body step of the EM-guided symmetry modeling procedure described previously (Bergeron et al., 2013). Briefly, the individual domains were manually placed in the corresponding region of the EM map, and 1,000 rigid-body decoy models were generated with imposed symmetry and with a restraint for fit into the FliF EM map density, at 22 Å resolution. The obtained models were isolated with the Cluster procedure in Rosetta, and scores calculated with the Score procedure using the lowest-energy model as a template for RMS calculations.

The flags used for the modeling procedure are listed in the Supplementary Methods section.

Results and Discussion

Characterization of the FliF domain organization

In order to identify conserved features, I gathered FliF sequences from a number of human pathogens, spanning gram-negative, gram-positive and spirochete bacteria. Most of the FliF sequences are similar in length (∼560 amino acids), but show limited sequence conservation (Table 1). I next used sequence analysis servers to predict secondary structure elements and other structural and functional features. Two transmembrane (TM) helices are predicted, between residues 20–45 and residues 445–470, and a secretion signal peptide targeting it for secretion and inner-membrane localization (Kudva et al., 2013) is predicted at the N-terminus. All FliF sequences show a very similar secondary sequence prediction pattern (Fig. S1), and similarity to SctJ was identified for residues 50–220 in all sequences (hereafter referred to as the SctJ homology domain), but no structural homologues were found for residues 220–440 (FliF-specific domain). Within this region, residues 305–360 of the FliF-specific domain are predicted unstructured in all sequences.

Table 1 Sequence identity between FliF orthologues used in the multiple sequence alignment shown in Fig. 2.

The sequence identity between each pair is indicated, as calculated with the Needleman–Wunsch algorithm.

	Salmonella	E.coli	Yersinia	Bordetella	Pseudomonas	Legionella	Helicobacter	Campylobacter	Listeria	Streptococcus	Vibrio	Bacillus	Clostridium	Treponema	
Salmonella	100.0	85.9	62.4	45.1	34.1	34.7	30.7	29.4	23.6	21.7	27.9	24.6	26.2	21.5	
E.coli		100.0	62.6	46.2	33.7	34.7	31.4	29.6	22.0	22.0	27.7	25.1	25.0	22.2	
Yersinia			100.0	49.2	35.5	34.5	29.6	28.1	21.2	21.1	26.6	24.2	23.9	19.5	
Bordetella				100.0	35.9	35.1	29.6	26.3	22.9	21.0	27.5	23.2	22.0	21.6	
Pseudomonas					100.0	38.1	28.4	29.0	20.3	20.3	31.0	20.0	22.2	20.3	
Legionella						100.0	28.8	26.5	20.2	20.2	29.3	21.2	25.3	20.2	
Helicobacter							100.0	43.2	23.7	21.9	23.3	23.8	28.3	24.2	
Campylobacter								100.0	22.8	21.2	26.1	23.1	23.3	25.0	
Listeria									100.0	37.4	19.8	22.3	24.5	20.5	
Streptococcus										100.0	20.5	22.4	24.4	21.3	
Vibrio											100.0	19.1	21.8	18.6	
Bacillus												100.0	24.6	23.2	
Clostridium													100.0	20.7	
Treponema														100.0	

I then generated a multiple alignment of all FliF sequences, and used it to map the predicted secondary structure and identified domains (Fig. 2). This further illustrates the conserved domain organization in all FliF orthologues. The two RBMs of the SctJ homology domain (labeled RBM1 and RBM2) are well conserved, as is the linker L1 between these. It has been shown that in SctJ this linker plays a role in ring assembly (Bergeron et al., 2015), suggesting that this may also the case in FliF. In contrast, the linker region L2, separating the SctJ homology domain to the FliF-specific domain, is highly variable.

Figure 2 Domain organization of FliF.

Multiple sequence alignment of FliF sequences from various human pathogens (S. typhimurium, Escherichia coli, Yersinia pestis, Bordetella pertussis, Pseudomonas aeruginosa, Legionella pneumophilia, Helicobacter pylori, Campylobacter jejuni, Listeria monocytogenes, Streptococcus pneumonia, Vibrio cholerae, Bacillus subtilis, Clostridium difficile, Treponema palladium). Conserved residues are in red box, similar residues are in red characters. Identified domains are highlighted in colored boxes, with the TM helices in yellow, the FliG-binding domain in green, the signal sequence in purple, the SctJ homology domain in blue and the FliF-specific domain in orange. The predicted secondary structure elements for the S. typhimurium FliF are in blue at the top.

The Chlamydiacae FliF-FliG fusion protein

While the FliF domain organization described above was found in most FliF sequences, one notable exception was identified, for the Chlamydiaceae family, where the FliF sequence is notably shorter (∼330 amino acids). Sequence analysis and multiple sequence alignment from all available Chlamydiaceae homologues (Fig. 3A) revealed that the N-terminal signal sequence and the two TM (residues 16–33 and 250–275) are present, but the periplasmic region is significantly shorter. Residues 60–145 show some sequence similarity to SctJ, but only encompassing RBM2. No structural homologues could be identified for residues 165–235, however the predicted secondary structure matches that of the canonical RBM.

Figure 3 The Chlamydia FliF orthologue has unusual domain architecture.

(A) Multiple sequence alignment of FliF sequences from the Chlamydiacae family (C. trachomatis, C. muridarum, C. suis from the genus Chlamydia, and C. psittaci, C. abortus, C. felis, C. caviae, C. ibidis, C. pneumonia, and C. pecorum from the genus Chlamydophila. Labeling is as in figure A, with the secondary structure prediction of the C. trachomatis orthologue shown at the top. (B) Schematic representation of FliF and its interaction with FliG (top), and of the Chlamydia FliF-FliG fusion (bottom).

In most orthologues, the C-terminus cytosolic region of FliF (residues 520–560) binds to the protein FliG (Levenson, Zhou & Dahlquist, 2012; Ogawa et al., 2015), a ∼37 KDa protein possessing three domains, labeled N, M and C. FliF interacts with domain N, while both M and C bind to the C-ring component FliM (Brown et al., 2007; Minamino et al., 2011), as illustrated on Fig. 3B. However, a sequence similarity search revealed that in the Chlamydia FliF, the cytosolic region is actually homologous to the M region of FliG (not shown), revealing that in this species the protein is in fact a FliF-FliG fusion. Chlamydia is not thought be a flagellated bacterium, and possesses only a few flagellar genes, namely FliF, FliL and FlhA. It does however possess a functional injectisome that is essential for virulence (Peters et al., 2007), and it has been shown that Chlamdia flagellar proteins interact with components of its injectisome (Stone et al., 2010). FliM (and FliN, another C-ring component) is homologous to the Chlamydia injectisome component SctQ (Notti et al., 2015). It is therefore possible that the FliF-FliG fusion interacts with SctQ (Fig. 3B). While the Chlamydia injectisome possesses both SctJ and SctD homologues (Nans et al., 2015), expression of fliF was shown to be concomitant with that of the injectisome (Hefty & Stephens, 2007). It remains to be tested experimentally if FliF is included in the Chlamydia injectisome.

Interestingly, a FliF-FliG fusion is not entirely unprecedented, as two artificial FliF-FliG fusions have been reported in S. typhimurium (Francis et al., 1992; Thomas, Morgan & DeRosier, 2001). In both cases, the fusion does not impair flagellum assembly and rotation, but induces a bias in the rotation direction. Since it is not known if the Chlamydia FliF-FliG fusion protein is part of a proto-flagellum, or contributes to the injectisome, the implications for this observation is not clear, but suggests that such a fusion protein may be functional.

Structural modeling of the SctJ homology domain

Structures of the periplasmic domains of both SctJ from EPEC (Crepin et al., 2005; Yip et al., 2005) and S. typhimurium (Bergeron et al., 2015) named EscJ and PrhK respectively, have been reported. Exploiting this information, a structural model for FliF was generated using the prototypical S. typhimurium FliF sequence. Despite the predicted structural homology, the sequence conservation between FliF and SctJ is low (Fig. 4A). I therefore employed the secondary structure alignment-based server Phyre (Kelley et al., 2015) for modeling the SctJ homology domain, spanning residues 50–221. However, the relative orientation of the two RBMs in FliF is not known, and may differ from that of SctJ. I therefore modeled the two RBMs independently, and refined the obtained models with Rosetta (see ‘Materials and Methods’ for details). As shown on Figs. 4B and 4C, both RBMs converged to a local energy minimum within an RMSD of 1 Å to the lowest-energy model in the refinement procedure. The models possess good geometry (Table 2), and their overall architecture (Figs. 4D and 4E) is expectedly similar to that of SctJ.

Figure 4 Modeling of the S. typhimurium FliF SctJ homology domain.

(A) Sequence alignment of the periplasmid domains from the EPEC and S. typhimurium SctJ homologues, EscJ and PrgK, with that of the SctJ homology domain of FliF (residues 50–221). Secondary structure elements for EscJ (PDB ID: 1YJ7) and PrgK (PDB ID: 3J6D) are shown at the top, in blue and green respectively. (B) and (C) Energy plot for the refinement of the FliF RBM1 and RBM2. The RMSD values are computed for all atoms, relative to the lowest-energy model. (D) and (E) Cartoon representation of the lowest-energy models for the FliF RBM1 and RBM2, with rainbow coloring indicating N- to C-termini.

Table 2 Geometry validation scores for the structural models of the three FliF RBMs.

Geometry parameters for each atomic models, obtained with the Protein Structure Validation Suite.

	RBM1	RBM2	RBM3	
Ramachandran favored (%)	98.0	97.8	98.9	
Ramachandran allowed (%)	0.0	1.1	1.1	
Ramachandran disallowed (%)	2.0	1.1	0.0	
Verify3D score	0.46	0.38	0.31	
Procheck score	0.39	0.26	0.20	
MolProbity Clashscore	1.25	4.22	13.10	
Close contacts	0	0	0	
RMSD bond angle (Å)	1.5	2.6	1.8	
RMSD bond length (°)	0.010	0.013	0.023	

The FliF-specific domain is a RBM

I next sought to generate a structural model for the FliF-specific domain (residues 228–443). As mentioned above, secondary structure prediction indicated that residues 228–309 and residues 356–443 possess defined secondary structure, while residues 310–355 are predicted as intrinsically disordered (Fig. 2). I therefore hypothesized that the FliF-specific region consists of two globular domains (D1 and D2) separated by a flexible linker. I then attempted to identify structural homologues to these two regions, using the Threading server PsiPred (Buchan et al., 2013). Surprisingly, D1 is predicted to have structural homology to SctJ, although only for the first two secondary structure elements (residues 229–268, Fig. S2A). Similarly, D2 was identified to have structural homology to the sporulation complex protein SpoIIIAH, which also possesses a RBM fold and has been proposed to oligomerize into ring structures (Levdikov et al., 2012; Meisner et al., 2012) (Fig. S2B). However, the structural homology was limited to the last three secondary structure elements (residues 386–436). Based on these observations, I postulated that the FliF-specific domain is a “split” RBM that possesses a large insert in the loop between the first and second strands (Fig. 5A). A secondary structure similarity search for the FliF-specific domain with this insert removed (FliF228 –443Δ274–378) confirmed overall fold similarity to both SctJ and SpoIIIAH (Fig. S2C).

Figure 5 Modeling of the FliF-specific domain.

(A) Sequence alignment of the EscJ RBM2, PrgK RBM2, SpoIIIAG and SpoIIIAH, with that of the FliF-specific domain (residues 228–439). Secondary structure elements for EscJ (PDB ID: 1YJ7) and SpoIIIAH (PDB ID: 3UZ0) are shown at the top, in blue and green respectively. The location of the insert ion FliF and SpoIIIAG is indicated. (B) Energy plot for the refinement of the FliF RBM3. The RMSD values are computed for all atoms, relative to the lowest-energy model. (C) Cartoon representation of the lowest-energy model for the FliF RBM3, with rainbow coloring indicating N- to C-termini. The location of the insert is indicated.

A “split” RBM is not unprecedented, as it is also predicted in the sporulation complex component SpoIIIAG (Fig. 5A). In both proteins an insert is predicted between the first and second strand of the RBM, with four putative β-strands in the inserted domain. This insert is large enough to accommodate a small globular domain; alternatively, it is possible that the insert adopts an extended loop conformation stabilized by β-strands.

In order to build an atomic model for the FliF RBM3, the SctJ-derived model for FliF230 –275 was combined with the SpoIIIAH-derived model for FliF380 –440, and the resulting structural model was refined with Rosetta. As shown on Fig. 5B, the RMSD to the lowest energy model is significantly higher than for the RBM1 and RBM2 models (around 3.5 Å for most decoys), which is perhaps expected, as the starting model was a presumably poorer, composite model. Nevertheless there is a clear energy funnel, indicative that the modeling procedure is converging. The obtained FliF RBM3 model, shown on Fig. 5C, possesses the canonical RBM fold, with good overall geometry (Table 2) and an elongated architecture similar to the RBM2 of SctJ and SpoIIIAH, with the insert between strands 1 and 2 located on one side of the structure.

Localization of the FliF domains

Previous EM studies have shown that in isolation, FliF forms a doughnut-shaped oligomer with two side rings corresponding to the MS rings seen in the intact basal body (Suzuki et al., 1998; Thomas et al., 2006). The FliG-binding domain forms the cytoplasmic M ring, but little is known about the organization of the periplasmic domains. However, the organization of the injectisome basal body, and that of SctJ in particular, is well characterized (Schraidt & Marlovits, 2011; Bergeron et al., 2015). By comparing the EM reconstructions of the flagellar (Thomas et al., 2006) and injectisome (Schraidt & Marlovits, 2011) basal body complexes, localization of the FliF RBMs can be inferred. As shown on Fig. 6A, density attributed to the two globular domains of SctJ has clear equivalence in the flagellum, suggesting this location corresponds to the SctJ homology domain of FliF. EM map density for the flagellum S-ring corresponds to the injectisome protein SctD localization, and is also present in purified FliF (Suzuki et al., 1998), confirming that it is not attributed to another flagellar component. The periplasmic region of SctD is composed of three domains with an RBM fold (Spreter et al., 2009; Bergeron et al., 2013), including the region corresponding to FliF density, suggesting that in the flagellum, this density can be attributed to the FliF-specific domain, which also possesses an RBM fold.

Figure 6 Comparison of the flagellar and T3SS basal body.

(A) EM maps of the S. typhimurium T3SS basal body (EMBD ID: 1875) in black, overlaid on that of the S. typhimurium flagellum basal body (EMBD ID: 1887) in yellow. A close-up view of the S-ring region of the flagellum is shown on the left, with the SctJ/SctD ring model (PDB ID 3J6D) docked in the T3SS EM map. The SctJ RBM1 is in cyan and the RBM2 in blue, and the three SctD RBMs are in green. (B) Schematic representation of SctJ, SctD and FliF, with the TMs in yellow, and the RBMs colored as in (A) for SctJ and SctD. Corresponding domains are indicated in FliF.

Based on these observations, I propose that FliF is akin to a SctJ-SctD fusion (Fig. 6B), including the two RBMs of SctJ and its C-terminal TM helix, and the N-terminal TM and first RBM from SctD. This likely explains a major difference between flagellar basal body, where FliF assembles on its own in the inner membrane, and injectisome basal body, where co-expression of SctJ and SctD is required (Kimbrough & Miller, 2000).

Further sequence analysis will be required to decipher the exact evolutionary pathway leading to the emergence of two genes in the injectisome. It is however interesting to note that in the ancestral Myxococcus injectisome, the SctD homologue lacks a periplasmic domain, and only consists in the cytoplasmic FHA-fold domain found in all SctD homologues, Bergeron et al. (2013) followed by a predicted transmembrane domain. It is therefore likely that this gene was fused to a FliF duplication gene in subsequent injectisomes, leading to the SctD homologue including the FHA cytosolic domain and the three BRMs in the periplasm.

One can wonder how having two separate proteins is beneficial for the injectisome. It may allow for a more dynamic assembly/disassembly process. Indeed, most injectisomes are only transiently functional, and while the regulation of injectisome genes transcription is well characterized, the fate of the injectisome complex after its required function (such as, for example, the SPI-1 injectisome of S. typhimurium after insertion in the Salmonella-Containing Vacuole) is not known. It is possible that having two proteins allow for disassembly of the basal body and recycling of the corresponding proteins (Diepold & Wagner, 2014).

It is also interesting to note that unlike SctJ and FliF, SctD lacks a secretion signal at its N-terminus. Indeed, deletion of the complete N-terminus of the S. typhimurium SctD homologue allows the formation of intact basal body complexes (Bergeron et al., 2013). It remains to be shown how the periplasmic domain of this protein is translocated across the inner membrane, but co-export with SctJ is an intriguing possibility, and to my knowledge such co-secretion via the Sec pathway (through which both FliF and SctJ are thought to be translocated) is unprecedented.

Modeling of the FliF oligomer

I next exploited the domain localization proposed above, the structural models of the three FliF RBMs, and the previously determined EM map (Suzuki, Yonekura & Namba, 2004), to generate a structural model of the FliF monomer. To that end, I positioned all three domains so that their termini point towards the correct region of the map, and fitted into the EM map density with Chimera (Goddard, Huang & Ferrin, 2007). Figure 7A shows that RBM1 is located near the inner-membrane region, while RBM2 forms the neck of the structure. Finally, the RBM3 is located in the region of density forming the S-ring. The insert in RBM3 points towards the lumen of the ring, and could correspond to the gate density observed in the FliF EM structure (Suzuki et al., 1998; Suzuki, Yonekura & Namba, 2004).

Figure 7 Modeling of the FliF oligomer.

(A) Docking of the three FliF RBM models in the FliF EM map. The domains are colored as in figure 5B. (B) Energy plot for the EM-guided symmetry docking procedure of the FliF RBM3. The RMSDs are computed for backbone atoms of the entire modeled 24mer complex, relative to the lowest-energy model, and color-coded depending on the fit to EM map. Three clusters of low-energy models were identified (Cl-1, Cl-2 and Cl-3), with two adjacent molecules for each cluster shown in (C). The 25-mer radius axis is represented by a dotted line. (D) 25-mer model of the FliF periplasmic region, viewed from the top.

I next attempted to generate a model for the FliF oligomer. I applied a previously described EM-guided modeling procedure (Bergeron et al., 2013) to all three domains individually, using the FliF map as a restraint, and applying 24-, 25- or 26-fold symmetry. In most cases the procedure led to ring models that were too large for the EM map (not shown). This is unsurprising, as it had been reported previously that the procedure requires experimental structures (as opposed to homology models) to converge (Bergeron et al., 2013). The one exception was the modeling of the RBM3, using 25-fold symmetry. This procedure generated three distinct clusters of models with clear energy funnels and good fit to EM density (Fig. 7B). Close inspection of the clusters reveals that they correspond to similar oligomerization modes and use the same interface, but with a slightly different angle between the subunits (Fig. 7C). I therefore exploited this 25-mer mode for the RBM3 to generate a complete 25-mer model for FliF. The corresponding model, shown in Fig. 7D and included in the Supplemental Information, matches well with the cryo-EM structure, with only a few loop regions of RBM2 and RBM3 located outside of the density.

I emphasize that while only 25-fold symmetry led to a convergent model for RBM3, this should not be considered as evidence that FliF forms a 25-mer. Indeed, the limited amount of data used in the modeling procedure is not sufficient to generate an accurate structural model, and at the resolution of the EM map (∼24 Å) the difference between 25 and 26 subunits is indistinguishable. It is also possible that the overall diameter of the complex appears smaller due to a small error in the pixel size of the detector used, thus biasing the modeling process towards a lower oligomeric state. However, the converging energy landscape for RBM3, and fit to EM density for all three domains, suggest that the reported model may be exploiting the native general orientation and oligomerization interface for all three domains. Additional experimental data, including experimentally determined RBM1, RBM2 and RBM3 atomic structures, as well as a higher resolution EM map of FliF and of the intact flagellum basal body, will be required to further refine the FliF model.

Despite these significant limitations, several aspects of the FliF model are worth commenting on. Firstly, the oblong shape of RBM2 forms an angle close to perpendicular to the membrane plane (Fig. 7A). In contrast, this domain is almost to parallel to the membrane plane in SctJ (Yip et al., 2005; Schraidt & Marlovits, 2011; Bergeron et al., 2013). While the proposed orientation of FliF RBM2 is driven by the EM map density of the isolated FliF oligomer, it is possible that it may differ in the intact flagellum basal body, and correspond to a structural rearrangement upon recruitment of other flagellar components (rod and/or LP ring). Higher resolution EM reconstruction of the full flagellum basal body will be required to identify structural rearrangements associated with flagellar assembly.

It is similarly noteworthy that in the FliF model, RBM2 is located approximately 20 Å away from RBM1, unlike SctJ where the two domains form a direct interaction (Bergeron et al., 2015). The later domain arrangement was shown to be mediated by the linker region, with in particular a conserved phenylalanine residue playing essential role in oligomerization (Phe 72 in the S. typhimurium SctJ homologue PrgK). Interestingly, the linker between RBM1 and RBM2 is also well conserved in FliF (linker L1 in Fig. 2), with in particular a Phe found in most orthologues at position 121 (and replaced with a different large hydrophobic residue in some species), which could perform a similar role. It is therefore likely that this linker plays a role on FliF assembly.

Finally, it is interesting to note that in the FliF model, RBM1 and RBM3 are in close proximity, and would likely form direct interactions. Considering the limitations of the modeling procedure described above, it would be premature to use this model to identify residues that are part of the interaction interface. Nonetheless, continuous density in this region of the injectisome EM map (Schraidt & Marlovits, 2011) suggests that a direct interaction does exist between these two domains, thus supporting a similar interaction in FliF.

Conclusion

In this study I have presented evidence that the periplasmic region of most FliF orthologues consists of three globular domains possessing the canonical RBM motif. One exception is the Chlamydia FliF paralogue, which possesses only two of these, and has a FliG domain fusion at its C-terminus. By comparison with the injectisome basal body, I also propose the novel concept that FliF is akin to a fusion of SctJ and SctD. Finally I have combined this information to propose a model for the oligomeric arrangement of the periplasmic region of FliF. Further experimental validation will be required to confirm these observations, and to refine the FliF model.

These results shed new lights on the architecture and evolution of the flagellum MS ring. Specifically, the domain organization of FliF highlights similarities with the injectisome, but also with the bacterial sporulation complex. The proposed concept that FliF corresponds to a fusion of SctJ and SctD likely explains why FliF can assemble spontaneously, while SctJ and SctD require co-expression. In addition, the identification of a Chlamydia FliF-FliG fusion suggests that this may correspond to an ancestral complex.

Supplemental Information

Figure S1 Secondary structure prediction for three distant FliF orthologue

The secondary structure prediction generated by the PSIPRED server is shown for the S. typhiomurium, T. pallidum and B. subtilis FliF sequences.

Click here for additional data file.

Figure S2 Results of the Phyre searches for the FliF-specific domain

The top scoring template(s) identified by the threading server Phyre are shown for three different regions of FliF RBM3.

Click here for additional data file.

Supplementary Methods Supplementary Methods

Flags and commands used for the EM-guided symmetry modeling procedure—See Materials and Methods

Click here for additional data file.

Supplemental Information 1 FliF model

Atomic model of the FliF oligomeric ring.

Click here for additional data file.

I am grateful to Dr. Keiichi Namba and Dr. Hirofumi Suzuki for providing the FliF EM map. I thank Dr. Natalie Zeytuni and Dr. Morgan Bye for critical comments on the manuscript.

Additional Information and Declarations

Competing Interests

Author Contributions

Data Availability

The authors declare there are no competing interests.

Julien R. Bergeron conceived and designed the experiments, performed the experiments, analyzed the data, wrote the paper, prepared figures and/or tables.

The following information was supplied regarding data availability:

The FliF oligomer model is included in the Supplemental Information.

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
