# Peer review of "Structural modeling of the flagellum MS ring protein FliF reveals similarities to the type III secretion system and sporulation complex"

_PeerJ, doi:10.7717/peerj.1718_

## Round 0.1 · original submission · Major Revisions

Dear Julien,

Your work was reviewed by three "flagellar" colleagues who, like me, found it interesting. It will need re-reviewing upon revision and for that I'd be grateful if you could suggest a couple of relevant bioinformatics experts to me too, since that is not my strongest area of expertise. I am looking forward to receiving your revision.

Best,
Ariel.

Reviewer 1 ·

Basic reporting

In this work, the author performed structural modeling of the MS-ring protein FliF of the bacterial flagellum.
Using a purely bioinformatical approach, a hypothetical structure of FliF was modeled and docked onto existing EM maps of the flagellar basal body.

I am not an expert, but the homology modeling and docking appears to be solid. However, without any experimental validation I have a hard time to judge how relevant the proposed modeled structure of FliF is.
The most interesting part of the manuscript is the hypothesis that FliF is a fusion protein of PrgK and PrgH. The author hints at important consequences for the evolution of the flagellum, but unfortunately does not elaborate here. If the hypothesis is correct that the flagellum is phylogenetically the ancestor of the injectisome, then a primordial FliF would have needed to split-up into the contemporary PrgK and PrgH proteins.

I’m not sure if evolution works like that, but this could be discussed in more detail (intuitively, I thought it would be more likely to produce a fusion protein of two existing proteins – in addition, prgK and prgH are not directly located next to each other on the chromosome of Salmonella).

The article itself is nicely written. I have some comments and several general misconceptions that need to be corrected.

1) Abstract, lines 25-26: First, only parts of the export apparatus are located in the cytosol (the ATPase complex). Major components are membrane-embedded. Second, it appears that the author has a misunderstanding about the definition of “stator” and “rotor” components of the flagellum (see also below).

2) Abstract, lines 28-29 and Introduction lines 57-62: I’m not sure about the statement that assembly of the flagellum is initiated by formation of the MS-ring. It is clear that formation of the MS-ring is needed for flagellum assembly to proceed and also that FliF can oligomerize in the absence of any other components (except FliG). But to my knowledge it has not been shown conclusively that formation of other components of the membrane-embedded export apparatus is not an independent event. This certainly seems to be the case for the injectisome export apparatus (see Wagner, S. et al. Proc. Natl. Acad. Sci. 107, 17745–17750 (2010) and Diepold, A. & Wagner, S. FEMS Microbiol. Rev. (2014). doi:10.1111/1574-6976.12061)

3) Introduction, line 54: should read “Salmonella Typhimurium” or “Salmonella enterica serovar Typhimurium”. Please check spelling of “Salmonella Typhimurium” throughout the manuscript

4) Introduction, lines 60-61: This is incorrect. The hook protein is not exported “to the cytoplasm” but rather outside of the cell and in addition, the export apparatus exports many more proteins (e.g. components of the rod, the rod cap and hook cap and later components of the filament, as well as regulatory proteins).

5) Introduction, line 67 and others: The flagellum includes rotary and stationary parts. MotAB, the motor-force generators are attached to the peptidoglycan and thus called the stator complex, whereas the C-ring rotates and thus is called the rotor.

6) Introduction, lines 71-75: The term “the type III secretion system” is a misconception. Type III secretion systems (T3SS) are the membrane and associated cytoplasmic proteins that form the protein export apparatus of both the flagellum and needle-complex or injectisome (see also Desvaux, M. et al Trends Microbiol 14, 157–160 (2006)).

7) Results, lines 181-186: Does Chlamydia also contain a PrgK homolog or is perhaps the FliF-FliG fusion protein incorporated into the injectisome?

Experimental design

see above

Validity of the findings

see above

Additional comments

see above

Reviewer 2 ·

Basic reporting

There are basic misunderstandings and wrong descriptions about the flagellar system in this manuscript. Probably the author confuses the stator with the rotor of the bacterial flagellar motor. The stator is a complex of membrane proteins, MotA and MotB, and not in the cytosol (line 26 and 51), but is embedded in the cell membrane. The stator is not a part of the secretion complex (line 60), and not recruited by the MS ring (line 59). FliG and FliM are components of the C-ring and the C-ring is not a stator component (line 67 and 176), but a part of the rotor.

Other mistakes are shown below.
line 54-56
The hook length of the Salmonella typhymurium flagellum is well controlled to 55nm. The flagellar filament is much longer than “several um”, typically more than 10 um for Salmonella.

line 60-61
The secretion complex exports the flagellar axial proteins to the central channel of the growing flagellum, not to the cytoplasm.

line 178
“revealed that the in” -> “revealed that in”

Experimental design

As describe in the “validity of the findings”, more evaluation of the reliability of the FliF ring model (line 263-288) is needed.

Validity of the findings

The evaluation of the reliability of the FliF ring model (263-288) is poor. From the previous many EM analyses and the rotational step measurement, the 26 fold symmetry is the currently most accepted model. However, the author failed to build the 24- and 26-fold symmetry models, and only succeeded in construction of the 25-fold symmetry model. Therefore more careful evaluation is required to justify the poposed model.

The author claims that his finding that FliF is akin to a fusion of PrgK and PrgH is an “unexpected observation” (line 98) and a “novel concept” (line 294-295). However, the concept is not new. PrgK and PrgH have been expected to be a counter part of FliF in T3SS (PNAS, 97:11008–11013 (2000), Cold Spring Harb Perspect Biol. 2(11):a000299 (2010), BBA, 181–206 (2004), Current Opinion in Structural Biology, 18:258–266 (2008)). The author clearly indicated the relation between FliF and the two T3SS proteins, PrgK and PrgH, from the careful sequence study, but the concept itself has already been discussed more than 10 years.

The description fro line 187 to 190 is misleading. The author argues that the FliF-G fusion is not unprecedented. However, the two mutants are artificial mutants fused the N-domain of FliG to the C-terminal domain of FliF. Thus it is not appropriate for the discussion.

The author uses “structural homology” (line 215) and “structural similarity (line 217 and 219), but the structure of FliF is not solved. What do they mean, 2D-structure or primary structure? Clarify them.

·

Basic reporting

The article by Bergeron on the structural modeling of the flagellum MS ring protein FliF is clearly written and logically built up. Results and Discussion sections are fused in this manuscript, which makes sense for this theoretical study. Figures are relevant for the content of the article but resolution can be improved as indicated below.

Minor comments:
Line 30: suggesting rather than demonstrating
Lines 76, 77: exchange word “homology” to “identity”. “Sequence homology” does not exist. Either the protein is homologous or not but this cannot be gradual.
Line 87: homolgous - homologous
Line 91: demonstrate?
Line 178: that the in the… needs resolution
Line 182: Chlamidia - Chlamydia
Line 198: typhymurium - typhimurium
Line 291: This theoretical study did not really “demonstrate” anything. I suggest using “presented evidence” or something similar less strong.
Line 292: Chlamydia in italic letters
Figure legend 7: “modls” - models
Figure 1: I suggest not to color code the flagellar rod and the injectisome inner rod both in brown as it has not been shown that they are functionally related.
Figure 5A: Letters are too small.
Figure 5B: Resolution of the screen shot is not good, in particular in the print out. Needs improvement.
Figure 7B: axis labeling is missing.
Figure S2AB: Resolution of the screen shot is not good and needs improvement.

Experimental design

Although it is clear from the text of the article why the presented work was performed, the research question is never clearly defined. This can be improved in the abstract and introduction section.
The methodology used and the technical standard seem to be according to the status quo of the field, although it is not the reviewers expertise to judge this sufficiently.

Validity of the findings

The data were obtained by theoretical approaches, measures of validity like rmsd and energy values were presented. Clearly, the discussion of placement of the identified domains in the basal body and of stoichiometry needs to be experimentaly validated but is beyond the scope of the presented study. It would be good, however, if the limitations of modeling the localization and oligomerization be discussed to a greater extent.

Additional comments

No comments

---

## Round 0.2 · accepted · Accept

Hi Julien,

I am pleased to accept your manuscript for PeerJ pending integration of the very minor revisions to the text that the three experts still suggests. Otherwise, very well done and thanks for submitting your work to PeerJ.

Best,
Ariel.

Reviewer 1 ·

Basic reporting

No Comments

Experimental design

No Comments

Validity of the findings

No Comments

Additional comments

The author has sufficiently addressed my major comments in the present revision.

I have the following minor points that should be corrected before publication:

1) My original comment no. 3 has not been addressed properly. “Typhimurium” is a serovar and should not be written in italics. The correct notation is “*italics*Salmonella*/italics* Typhimurium” or “*italics*Salmonella enterica*/italics* serovar Typhimurium”. Please adjust spelling of “Salmonella Typhimurium” throughout the manuscript.

2) line 58: cite ref: Hirano, T., Yamaguchi, S., Oosawa, K. & Aizawa, S. J Bacteriol 176, 5439–5449 (1994).

3) line 282: Myxococcus should be italic

4) line: 291: Salmonella should be italic

Reviewer 2 ·

Basic reporting

The revised manuscript has greatly improved. I think it’s almost ready for publication, but the author still seems to misunderstand the bacterial flagella rotor. The rotor is composed of the MS-ring (FliF) and the C-ring. The flagellar basal body includes the LP-ring, the MS-ring, the rod and also the cytoplasmic C-ring. So I recommend to revise the following sentences before publication.

line 25-26
“The flagellum consists of the export…”
line 56-58
“The membrane-embedded structure that…”
line 63-64
“recruits the rotor” -> “recruits the C-ring”

Experimental design

No Comments

Validity of the findings

No Comments

·

Basic reporting

Dr. Bergeron has satisfyingly responded to the reviewer's comments and made suggested changes accordingly. In particular I support his use of the unified nomenclature of the injectisome in this revised version of the manuscript and acknowledge the much improved quality of the figures.

Specific comments:
Lines 294-300: I cannot follow the argumentation in this point. SctD clearly contains a transmembrane segment, which serves as a secretion signal. Transmembrane proteins typically do not contain additional cleavable signal sequences in prokaryotes (with exceptions like the flagellar FliP), so SctD is not the exception but the rule. SctRSTUV altogether do not contain additional signal sequences but just transmembrane segments as targeting and stop transfer signals, so the statement of the sentence starting in line 294 is wrong. I suggest to remove this paragraph of the manuscript.
Line 296: StcD should be SctD if not removed according to the comment above.

Experimental design

No new comments.

Validity of the findings

No new comments.

Additional comments

No comments